# Utility of cell-free DNA concentrations and illness severity scores to predict survival in critically ill neonatal foals

**Sarah Florence Colmer**[ID]◉, **Daniela Luethy**[ID]‡, **Michelle Abraham**‡, **Darko Stefanovski**, **Samuel David Hurcombe**[ID]*◉

Department of Clinical Sciences, The University of Pennsylvania School of Veterinary Medicine, New Bolton Center, Philadelphia, Pennsylvania, United States of America

◉ These authors contributed equally to this work.
‡ These authors also contributed equally to this work.
* hurcombe@vet.upenn.edu

**Data Availability Statement:** All relevant data are within the manuscript and its Supporting Information files.

## Abstract

Plasma cell-free DNA (cfDNA) levels have been associated with disease and survival status in septic humans and dogs. To date, studies investigating cfDNA levels in association with critical illness in foals are lacking. We hypothesized that cfDNA would be detectable in the plasma of foals, that septic and sick-nonseptic foals would have significantly higher cfDNA levels compared to healthy foals, and that increased cfDNA levels would be associated with non-survival. Animals used include 80 foals of 10 days of age or less admitted to a tertiary referral center between January and July, 2020 were stratified into three categories: healthy (n = 34), sick non-septic (n = 11) and septic (n = 35) based on specific criteria. This was a prospective clinical study. Blood was collected from critically ill foals at admission or born in hospital for cfDNA quantification and blood culture. Previously published sepsis score (SS) and neonatal SIRS score (NSIRS) were also calculated. SS, NSIRS, blood culture status and cfDNA concentrations were evaluated to predict survival. Continuous variables between groups were compared using Kruskal-Wallis ANOVA with Dunn's post hoc test. Comparisons between two groups were assessed using the Mann-Whitney $U$-test or Spearman rank for correlations. The performance of cfDNA, sepsis score and NSIRS score to predict survival was assessed by receiver operator characteristic (ROC) curve analysis including area under the curve, sensitivity and specificity using cutoffs. Plasma cfDNA was detectable in all foals. No significant differences in cfDNA concentration were detected between healthy foals and septic foals ($P$ = 0.65) or healthy foals and sick non-septic foals ($P$ = 0.88). There was no significant association between cfDNA and culture status, SS, NSIRS or foal survival. SS (AUC 0.85) and NSIRS (AUC 0.83) were superior to cfDNA (AUC 0.64) in predicting survival. Although cfDNA was detectable in foal plasma, it offers negligible utility to diagnose sepsis or predict survival in critical illness in neonates.

**Funding:** This work was funded by the ACVIM 2019-2020 Resident Research Grant (grant number 775360) as well as the Raymond Firestone Trust Grant / New Bolton Center internal funding (grant number 461612-5875). The funders had no role in study design, data collection and analysis, decision to publish, or preparation of the manuscript.

**Competing interests:** The authors have declared that no competing interests exist.

## Introduction

Sepsis is the most common cause of neonatal foal mortality and is broadly defined by an overwhelming systemic inflammatory response syndrome triggered by infectious organisms [1]. A myriad of factors are involved in the pathophysiology of neonatal sepsis, including maternal disease, parturition complications and management practices [2]. The clinical signs observed in sepsis overlap with many other non-infectious, life-threatening conditions, and due to this ambiguity, diagnosis proves difficult until the disease become severe [3]. Recent studies show septic foals still have a high mortality rate of 30–50% despite improved survival with medical advances [4, 5].

The gold standard for the detection of bacteremia is blood culture which exhibits low sensitivity in neonates at approximately 61.9% [6]. False negative blood cultures in septic foals as well as positive blood cultures documented in healthy foals, further complicate the interpretation of results [7, 8]. Blood cultures regularly require up to one week before they can be considered diagnostic. Scoring systems were developed in the 1980s using blood culture and clinical findings to expedite and predict a diagnosis of sepsis in foals. These scoring systems incorporated both subjective and objective clinical and clinicopathologic findings and have yielded varying diagnostic accuracy suggesting variation within at-risk foal populations at different times and locations [9].

Recent studies in human and veterinary medicine have shown the prognostic value of cell-free DNA (cfDNA) in the blood of septic patients [10–12]. cfDNA is proposed to be released from various cell types during apoptosis and necrosis, and thought to be involved in host defense systems with the intent to trap and kill bacteria [13]. Previously, quantification of cfDNA was a labor- and cost-intensive process involving DNA isolation, extraction, gel electrophoresis and real-time PCR [14]. More recently, rapid and direct fluorescence assays have been developed that use nucleotide-binding fluorophores to identify cfDNA directly within biological fluids [14–16].

In humans, cfDNA concentrations have proven to be sensitive and specific for poor outcomes [17, 18]. One study analyzing 255 patients with severe disease revealed that admission cfDNA concentrations were higher in ICU non-survivors than in survivors [18]. A 2017 study assessing cfDNA concentrations in dogs using fluorometric assays found significantly higher concentrations in those presenting to an emergency service with sepsis or non-septic SIRS compared to healthy dogs [19]. Though the body of cfDNA research is growing in both human and veterinary medicine, there are no studies documenting its utility in equine neonatal sepsis to the authors' knowledge.

We believe that cfDNA may have diagnostic and prognostic utility for diagnosing sepsis in critically ill foals. We hypothesized that cfDNA would be detectable in the plasma of newborn foals and that the magnitude of cfDNA concentrations would be proportional to the severity of illness. Our specific aims were to measure and compare cfDNA without DNA extraction from healthy, sick non-septic and septic foals at admission to or born in the hospital. Further, we wanted to determine if there is an association between cfDNA concentrations and detectable bacteremia as well as previously published critical illness scoring systems, namely sepsis score and neonatal systemic inflammatory syndrome score [20]. Finally, we sought to determine if cfDNA concentrations and illness severity score could predict survival in critically ill foals.

## Materials and methods

### Study population

The study population consisted of foals ≤ 10 days of age who were either admitted to or born at the University of Pennsylvania School of Veterinary Medicine New Bolton Center between

January and July 2020. Inclusion criteria required the results of a blood culture performed within 24 hours of admission, documentation of either discharge from the hospital or non-survival, and recorded physical examination and clinicopathological parameters for the implementation of the updated sepsis scoring system and neonatal SIRS (NSIRS) scoring system published by Wong et al. in 2018 [20]. Foals were excluded if they were > 10 days of age and did not have a blood culture performed during hospitalization. All experimental procedures were approved by the University of Pennsylvania Institutional Animal Care and Use Committee (IACUC) under protocol # 806836, and undertaken with informed client consent. The IACUC specifically reviewed and approved that the study would not influence treatment protocols of individual cases. Humane endpoints were not predefined as part of the study protocol. Efforts to minimize suffering and distress included the use of sedation (xylazine, detomidine, butorphanol) and use of analgesics (flunixin meglumine) when deemed appropriate. The definition of non-survivors in this study were foals who died or were euthanized due to grave medical prognosis in hospital. The determination of grave medical prognosis was made by the licensed attending clinician. Animal health and behavior were monitored every 1–6 hours based on level of criticality utilizing clinicians' judgement. Animals were all housed inside the intensive care unit which consisted of individual, temperature-controlled stalls for each mare/foal pair. Research staff responsible for animal care and handling were all licensed veterinarians and veterinary nurses.

## Classification

Foals were classified into 1 of 3 groups: septic, sick non-septic and healthy as based on previously published criteria [21, 22]. Foals were defined as septic if they fulfilled any of the following criteria: (1) positive blood culture, and/or (2) a sepsis score of $\geq 12$ [20]. Sick non-septic foals were defined as those with a negative blood culture and a sepsis score between 6 and 11. Healthy foals were defined as those with a negative blood culture and a sepsis score $\leq 5$.

Survival was defined as any foal born at or admitted to New Bolton Center who was discharged alive. Non-survival was defined as any foal born at or admitted to New Bolton Center who died or was euthanized due to grave medical prognosis. Foals euthanized due to financial constraints were excluded from data analysis.

## Data collection

Historical data including maternal health during pregnancy, foaling date and details of foaling (Cesarean section, dystocia, induction protocol) were obtained. Age at presentation, breed and sex were recorded. Physical examination findings including rectal temperature, heart rate, respiratory rate, the presence of swollen joints, diarrhea, petechiation, scleral injection, hypopyon, anterior uveitis, respiratory distress and the presence of seizures were recorded. The sepsis score (SS) and neonatal systemic inflammatory response syndrome score (NSIRS) were calculated for each foal [20]. Outcome was recorded as survival to hospital discharge or non-survival.

Clinicopathologic data collected included a compete blood count (CBC) (Element HT5, Heska, Loveland, CO), serum biochemistry (VITROS 350 Chemistry System, Ortho Clinical Diagnostics, Buckinghamshire, England), blood glucose concentration (Accu-Chek Performa glucometer, Roche, Indianapolis, IN), blood L-lactate concentration (Nova 8+ Electrolyte analyzer, Nova Biomedical, Waltham, MA), plasma fibrinogen concentration (ACL Elite, Instrumentation Laboratory, Bedford, MA), serum immunoglobulin G (IgG) concentration (DVM Rapid Test, MAI Animal Health, Elmwood, WI), blood culture (BD SEPTI-CHEK media bottles, BD Biosciences, Franklin Lakes, NJ) (VersaTREK Automated Microbial Detection

System, ThermoFischer Scientific, Pittsburgh, PA) and cfDNA concentration (Qubit 4 fluorometer, Fisher Scientific, Pittsburgh, PA and Qubit™ dsDNA HS Assay Kit, Fisher Scientific, Pittsburg, PA).

## Assay validation and pilot study

Pilot data was performed on thawed, previously frozen (-80˚C) equine plasma samples from 10 healthy foals without DNA extraction. Phosphate buffered saline was used as a negative control. Each sample was run in triplicate and cfDNA was determined from the standard curve generated by low and high concentration lambda (*E. coli* bacteriophage) DNA standards. The intraassay coefficient of variation (CV) was 7.6%.

For foals of the reported study, samples from all foals as well as the 10 healthy foals from the pilot study were run in triplicate. The intraassay CV was 3%. Using the cfDNA values from the 10 healthy foals, the interassay CV was 5.4%.

Manufacturer specifications report the intraassay CV for low concentrations of DNA (>0.5ng/mL) was ≤2%. Further, the fluorescence stability shows <2.5% drop after 40 readings within a DNA concentration range of 1 to 500 ng/mL.

## Sampling and processing

Approximately 25 mL of whole blood was obtained by jugular venipuncture using aseptic technique at the time of catheter placement or within 12 hours of admission or birth. Blood cultures were processed via inoculation (8mL of blood) and incubation of SEPTI-CHEK media bottles at 35˚C for up to 7 days and detection using the VersaTREK Automated microbial system. When turbidity was apparent or 7 days had elapsed, the broth was Gram-stained and subcultured using trypticase soy agar with 5% sheep's blood and MacConkey agar for aerobic isolation and using Brucella blood agar for anaerobic isolation.

Citrated plasma was collected for cfDNA quantification. Briefly, 2.7 mL of whole blood was collected into a citrate collection tube (BD Biosciences, Franklin Lakes, NJ) and centrifuged (Centritic Centrifuge, Fisher Scientific, Pittsburgh, PA) for 10 minutes at 1370 g within 30 minutes of collection. Plasma was then pipetted into polypropylene conical tubes. A small amount of plasma was left in the citrate tube so as not to disturb the buffy coat. Tubes were stored at -40˚C, which has previously been shown to preserve sample integrity for cfDNA analysis [23]. At the time of batch processing, samples were thawed by allowing them to come to room temperature (20-22˚C) and cfDNA quantification measures were performed in triplicate using a benchtop fluorometer (Qubit 4 fluorometer, Fisher Scientific, Pittsburgh, PA) and associated reagents (Qubit 1x dsDNA HS Assay Kit, Fisher Scientific, Pittsburgh, PA) in thin-walled polypropylene tubes (Qubit Assay Tubes, Invitrogen, Carlsbad, CA) according to manufacturer's instructions.

## Statistical analysis

Prior to initiating the study, sample size calculation was performed based on prior prevalence data on the reported proportions of critically ill foals who were bacteremic [24]. Given that cfDNA concentrations have been associated with bacteremia in critically ill humans [25, 26] and specific data regarding cfDNA in horses are lacking, we elected to use foal bacteremia data as a guide to determine sample size. With a power of 80% and significance set at $P < 0.05$, at least sixty (60) animals were deemed necessary for this prospective study. Data were assessed for normality using the Shapiro-Wilk statistic. Parametric data are reported as mean (standard deviation) and non-parametric data are represented as median and range unless otherwise stipulated. Continuous variables between groups were compared using Kruskal-Wallis

ANOVA with Dunn's post hoc test. Comparisons between two groups (blood culture positive and culture negative; survivor and non-survivor) were assessed using the Mann-Whitney *U*-test. Correlations between continuous variables were evaluated using Spearman rank test. The performance of cfDNA, sepsis score and NSIRS score to predict survival was assessed by receiver operator characteristic (ROC) curve analysis including area under the curve, sensitivity and specificity using cutoffs. Significance was set to $P < 0.05$. Analyses were performed using Prism Version 8.4.2 (GraphPad Software, La Jolla, CA) and STATA (STATA Corp LLC, College Station, TX).

## Results

### Study population demographics

A total of 80 foals were enrolled in the study between January and July, 2020. There were 34 healthy foals (43%), 35 septic foals (44%), and 11 sick non-septic foals (13%). The median age of all foals in the study at the time of sampling was 0 days of age (range 0 to 10 days). The median age different foal groups were 0 days (range 0 to 2) for healthy foals, 0 days (range 0 to 10) for sick non-septic foals and 0 days (range 0 to 4) for septic foals and not significantly different (P>0.99). Of the 80 foals, 45 were colts (56%) and 35 were fillies (44%). Breeds represented within the study population included Thoroughbred (n = 48), Standardbred (n = 18), Warmblood (n = 7) and 1 each of the following: Appaloosa, Arabian, Clydesdale, Friesian, Miniature Horse, Morgan and Pony. There was no significant difference in sex or breed between the 3 groups. A total of 69/80 (86%) foals survived to discharge, and 11/80 (14%) of foals were euthanized or died. For sick hospitalized foals (sick non-septic and septic foals), 35/46 (76%) survived to discharge with 11/46 (24%) non-survivors. There was no significant difference in age between survivors and non-survivors in this population (P = 0.47). Cause of death or euthanasia due to associated poor prognosis included: neonatal encephalopathy (n = 4), congenital cardiac malformation (n = 2), Lethal White mutation (n = 1), atresia coli (n = 1), placental insufficiency (n = 1), enterocolitis (n = 1) and septic peritonitis (n = 1).

Septic foals as defined using SS ≥ 12 and/or positive blood culture had a median sepsis score of 10 (range 0 to 29) at the time of presentation. The median NSIRS score for septic foals was 2 (range 0 to 5). In the septic group, the mortality rate was 26% (9/35), of which 2/9 (22%) died in hospital and 7/9 (78%) were euthanized. The most commonly diagnosed disease process/comorbidity at the time of admission in the septic foal group was neonatal encephalopathy (n = 12), followed by diarrhea (n = 5), failure of transfer of passive immunity (n = 4), dystocia (n = 3) and dysmaturity (n = 3). Additional diagnoses included colic (n = 1), dysphagia (n = 1), atresia coli (n = 1) and Lethal White Foal Syndrome (n = 1).

Healthy foals, as defined by a sepsis score ≤ 5 and negative blood culture had a median sepsis score of 2.5 (range 0 to 5) and median NSIRS score of 1 (0 to 2) at the time of presentation. The mortality rate was 0% (0/34). The most common reasons for presentation at the time of admission was birth as part of a standard foaling program package for low-risk mares (n = 11), mild neonatal encephalopathy (n = 8) and meconium impaction (n = 4). Three foals were presented as companions to their dams and two foals were presented for mild idiopathic dysphagia. Two foals were presented as part of a standard foaling program package for high-risk mares. Sick non-septic foals had a median sepsis score of 9 (range 6 to 11) and median NSIRS score of 1 (range 0 to 2) with a mortality rate of 22% (2/9). Both animals were euthanized due to grave prognosis. The most common reasons for presentation at the time of admission were dystocia (n = 3) and diarrhea/enterocolitis (n = 3).

## Blood culture results

A total of 80 blood cultures were taken from 80 foals within 2 hours of admission or birth. Of foals that had culturable bacteria (24/80 (30%)), 39 isolates were obtained. Of those isolates, 23/39 (59%) were Gram-positive and 16/39 (41%) were Gram-negative. The most common isolates were of the genus *Staphylococcus* (n = 7), *Bacillus* (n = 3), *Acinetobacter* (n = 3), *Escherichia* (n = 3) *Streptococcus* (n = 2), *Microbacterium* (n = 2), *Pseudomonas* (n = 2), *Stenotrophomonas* (n = 2), *Arthrobacter* (n = 2), *Pantoae* (n = 2) *and Enterococcus* (n = 2). Additional isolates (n = 1) included *Actinobacillus*, *Rhizobium*, *Curtobacterium*, *Corynebacterium*, *Klebsiella*, *Weissella*, and *Dietzia*. The mean NSIRS score of the bacteremic population was 1.64 (standard deviation = 1.15), and the mean NSIRS score of the non-bacteremic population was 1.26 (standard deviation = 1.21). The mean sepsis score of the bacteremic population was 8.29 (standard deviation = 7.01), whereas the mean sepsis score of the non-bacteremic septic population was 6.28 (standard deviation = 5.09).

For foals who died, 5/11 (45%) had a positive blood culture. All positive cultures grew Gram-positive organisms with a further 2/5 also culturing Gram-negative organisms. Cultured isolates included bacteria of the following genus: *Bacillus*, *Acinetobacter*, *Micrococcus*, *Staphylococcus*, *Dietzia*, *Klebsiella* and *Enterococcus*.

## cfDNA concentrations and foal classification

There were no differences in cfDNA concentrations between groups of foals (**Fig 1**; P>0.05). Median cfDNA concentrations were 279.72 ng/mL (range 121.33 ng/mL to 424.67 ng/mL) in

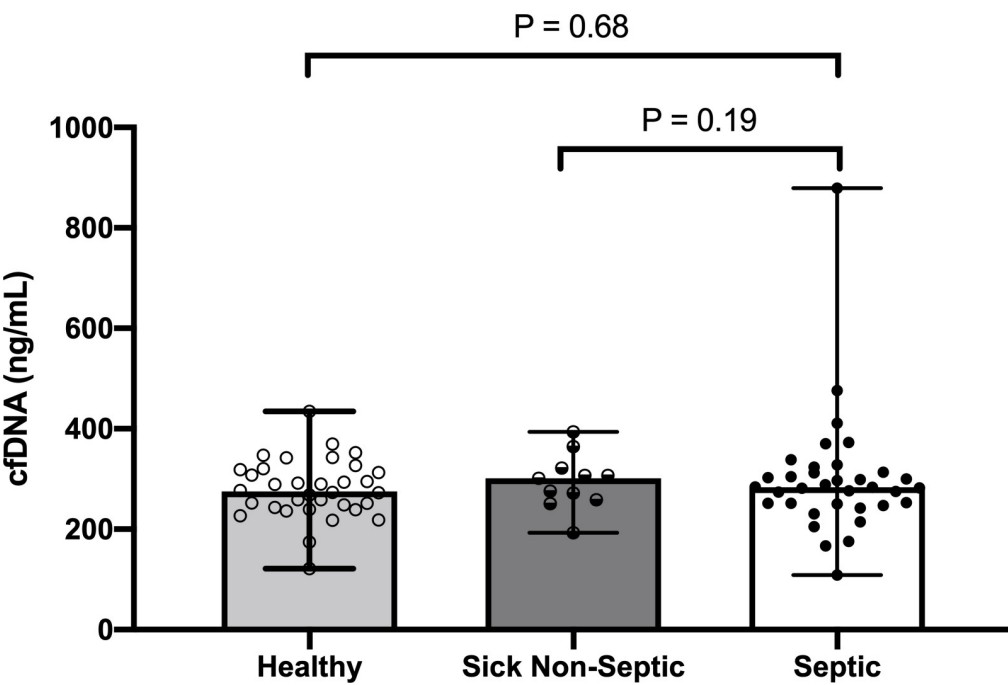

**Fig 1. Plasma cell-free DNA (cfDNA) concentrations in healthy foals (n = 34), sick non-septic foals (n = 11) and septic foals (n = 35) based on defined criteria (sepsis score and/or blood culture status).** Symbols represent values for individual foals with median and interquartile range error bars (P>0.05 between groups).

**Table 1. Cell-free DNA concentrations in 80 hospitalized neonatal foals based on sepsis score cut-off values alone (excluding blood culture status), including 44 healthy foals, 21 sick non-septic foals and 15 septic foals.**

| Foal Category | Median (range) cfDNA (ng/mL) | 95% Confidence Interval |
|---|---|---|
| **Healthy (SS ≤ 5) (n = 44)** | 290 (121 to 435) | 268.52 to 298.03 |
| **Sick Non-septic (SS 6–11) (n = 21)** | 274 (109 to 879) | 255.39 to 306.94 |
| **Septic (SS ≥ 12) (n = 15)** | 289 (176 to 476) | 188.19 to 317.92 |

Values expressed as median (range) and 95% confidence interval. SS = sepsis score.

healthy foals, 301.33 ng/mL (range 193–393.67 ng/mL) in sick non-septic foals and 299.15 ng/mL (166.67 to 879.33 ng/mL) in septic foals.

We further evaluated cfDNA concentrations based on SS alone to account for false positive blood cultures in healthy or non-septic foals and false negative blood cultures in septic foals. **Table 1** shows cfDNA concentrations between groups of foals based on SS cut-off criteria alone (septic ≥ 12, SNS 6–11, healthy ≤ 5) excluding blood culture results. cfDNA concentrations were not different between foals when classifying based on SS alone (healthy versus septic P = 0.37; sick non-septic versus septic P = 0.43).

The relationship between sepsis score and cfDNA concentration is illustrated in **Fig 2A** and was not significantly correlated (ρ = 0.07, P = 0.48). Further, in foals with SS ≥ 12 there was no significant correlation between cfDNA concentration and SS value (ρ = 0.25; P = 0.36). Similarly, there was no significant correlation between NSIRS score and cfDNA concentration (**Fig 2B**; ρ = 0.11; P = 0.54). There were no significant difference in median cfDNA concentrations between bacteremic foals (298.67 ng/mL, range 121.3 to 434.7 ng/mL) and non-bacteremic foals (275.33, range 109 to 879.3 ng/mL; P = 0.07; **Fig 3**).

## cfDNA concentrations and age

There was no correlation between cfDNA concentration and age in healthy foals (ρ = 0.16; P = 0.36) or overall for all foals (ρ = 0.13; P = 0.26). Further analysis evaluating the effect of age on cfDNA concentrations showed that for each day of age, there was a 1.72% increase in cfDNA concentration although not statistically significant (P = 0.13).

## cfDNA concentrations and critical illness scoring systems in predicting outcome

cfDNA concentrations between survivors and non-survivors are shown in **Fig 4**. There was no significant difference in cfDNA concentrations between foals that survived (median 286.88 ng/mL (range 121.33 to 879.33 ng/mL) and non-survivors (314.5 ng/mL, range 176.00 to 476.33; P = 0.16). Univariate logistic regression showed no association between survival and cfDNA concentration (P = 0.41). Receiver operating characteristic (ROC) curve analysis revealed poor utility for cfDNA to predict survival (AUC 0.64, P = 0.15; **Fig 5A**). ROC for sepsis score (**Fig 5B**) and neonatal SIRS score (**Fig 5C**) showed superior prediction to predict survival than cfDNA with an AUC for sepsis score of 0.85 (P<0.001) and for NSIRS score of 0.83 (P<0.001). **Table 2** shows cut-off values for cfDNA, SS and NSIRS to yield the optimal diagnostic accuracy for survival.

## Discussion

In the present study, cfDNA was detected in 100% of plasma samples obtained from this population of 80 foals admitted to a tertiary care facility. This finding supported our hypothesis that

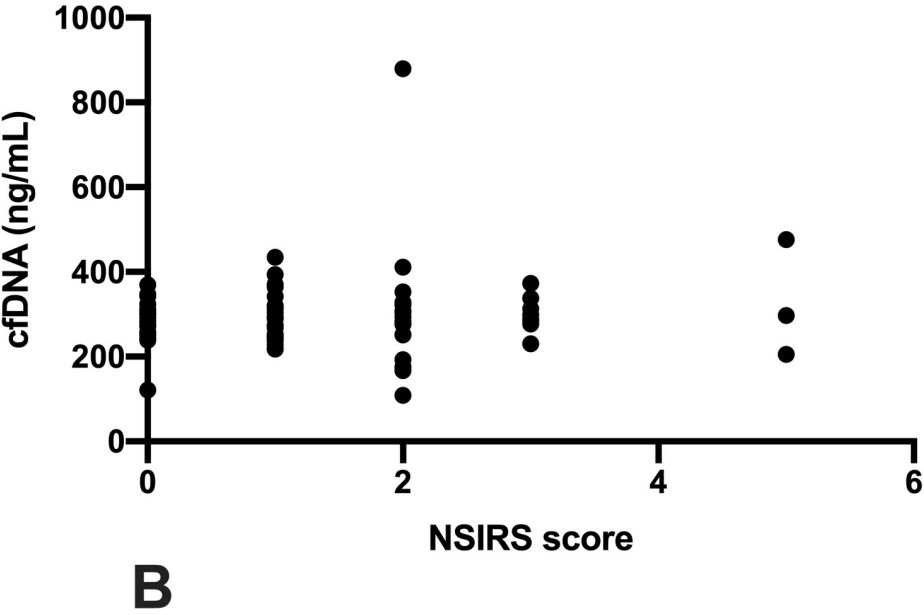

**Fig 2. Plasma cell-free DNA (cfDNA) concentrations and illness severity scores in 80 hospitalized foals.** 2A: cfDNA concentration and sepsis score (r = 0.07; P = 0.48). 2B: cfDNA concentration and NSIRS score (r = 0.07; P = 0.54). Symbols represent values for individual foals.

## cfDNA concentrations and blood culture status in foals

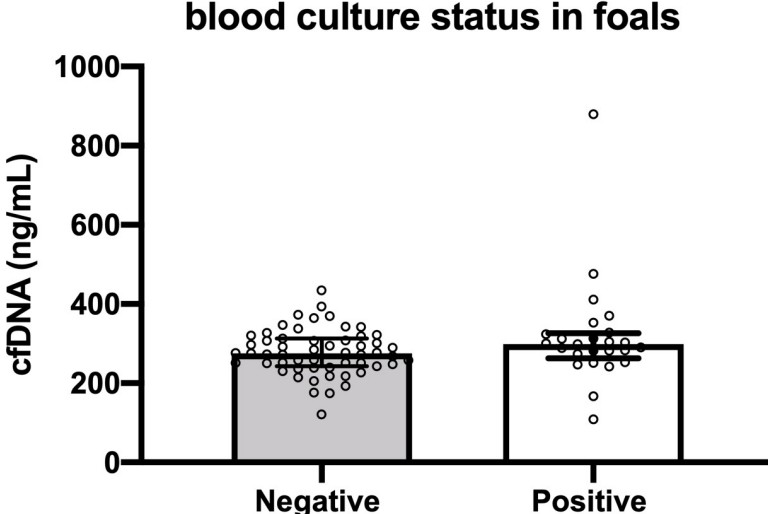

**Fig 3. Plasma cell-free DNA (cfDNA) concentrations in 80 hospitalized neonatal foals with positive blood culture (n = 24) and negative blood culture (n = 56).** Symbols represent values for individual foals with median and interquartile range error bars (P = 0.07).

cfDNA would be detectable in plasma samples obtained from the equine neonate. Though circulating plasma cell-free DNA has been detected in multiple veterinary species including canines [10, 19, 27, 28] and felines [29], this is the first study confirming its presence in neonatal foals. Results of this study demonstrate a lack of significance in circulating cfDNA concentrations between foal cohorts with varying illness severity. Further, cfDNA concentrations were not correlated with SS or NSIRS and we did not demonstrate an association between cfDNA and blood culture status or survival.

Recent studies in humans [11, 17, 18, 30] and dogs [10, 19] have illustrated the value of cfDNA as a potential biomarker for the detection of sepsis through its ability to discriminate

## cfDNA concentrations and survival status in foals

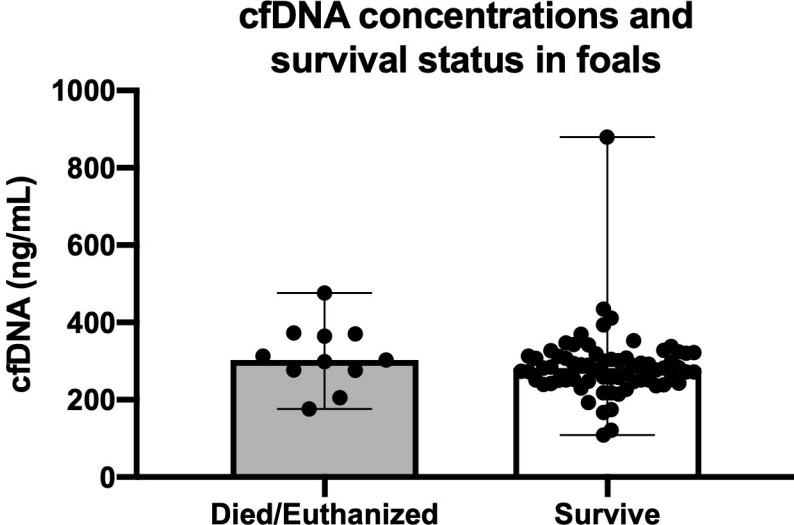

**Fig 4. Plasma cell-free DNA (cfDNA) concentrations in 80 hospitalized neonatal foals that survived (n = 69) and died (n = 11).** Symbols represent values for individual foals with median and interquartile range error bars (P = 0.16).

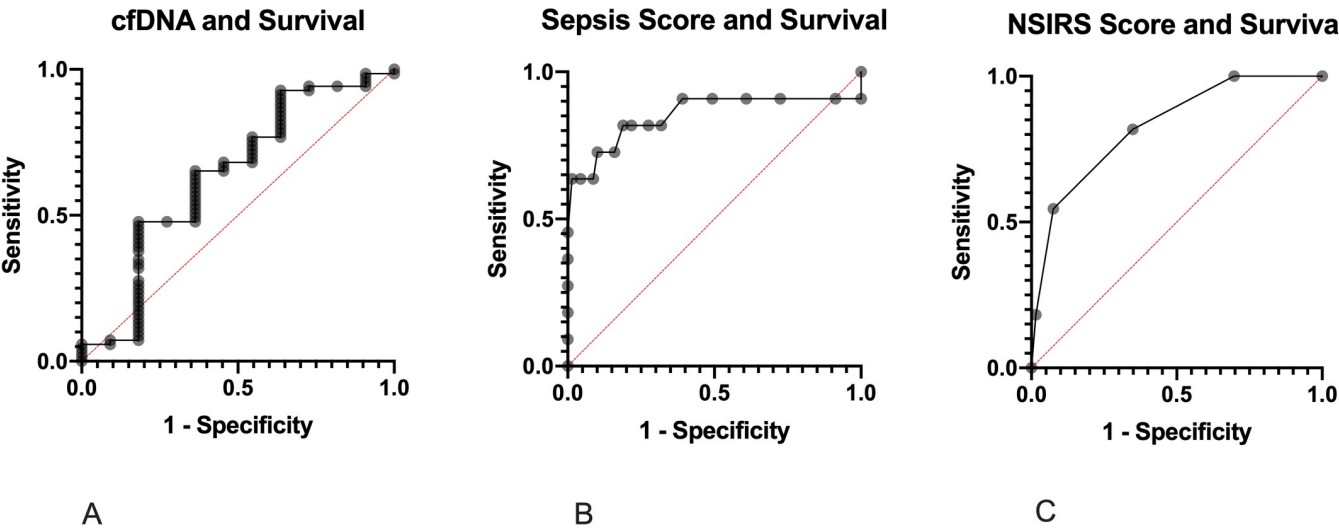

**Fig 5.** Receiver operator characteristic (ROC) curves for cfDNA (A), sepsis score (B) and NSIRS score (C) for predicting survival in 80 hospitalized neonatal foals.

septic and healthy patients. Of note, all of these studies have been performed on adult populations, differing from the age group of the findings reported here. In 2008, Tuaeva et al. investigated levels of cfDNA in the blood plasma of premature human neonates [31]. The ranges of cfDNA concentrations obtained were between 9.9 and 136.5 ng/mL [31] across all groups, significantly lower than ranges in other studies performed in adults. The relatively low concentrations of cfDNA in all foals of this study, regardless of illness severity, could be a function of species, disease state, immune state [32] and/or age. We observed a trend for increasing cfDNA concentrations with age and further prospective studies in equine neonates into adulthood would help elucidate the effect of age on circulating cfDNA concentrations in horses.

Although the source of circulating cell-free DNA is debatable, it is believed that circulating levels are related to cellular breakdown and active DNA-release mechanisms in host cells [33]. The concept of neutrophil extracellular traps, or NETs, involves the release of web-like scaffolds of cell-free DNA composed of extracellular chromatin which can enhance antimicrobial activity in the face of an infectious stimulus [34]. A model utilizing canine neutrophils has revealed that high-dose lipopolysaccharide stimulates the cells to undergo the production of NETs *in vitro* [35]. Investigating such mechanisms in the neonate, it is possible that our samples were obtained from an immune-competent but immune-naïve population based on age (10 days or less) where the degree of neutrophil activation, function or apoptosis is decreased compared to adult horses thereby limiting the amount of cell-free DNA released into circulation [36]. Including positive controls in the form of adult horses with well-documented

**Table 2. Diagnostic cut-off values for cfDNA, sepsis score (SS) and neonatal systemic inflammatory response syndrome (NSIRS) score to predict survival in 80 hospitalized neonatal foals.**

|  | AUC | 95% Confidence Interval | P value | Cut-off Value | Sensitivity | Specificity |
|---|---|---|---|---|---|---|
| **cfDNA (ng/mL)** | 0.64 | 0.4 to 0.8 | 0.15 | 297 | 64% | 65% |
| **Sepsis Score** | 0.85 | 0.67 to 0.99 | 0.0002 | 9.5 | 82% | 81% |
| **NSIRS Score** | 0.83 | 0.7 to 0.96 | 0.0005 | 2.5 | 93% | 55% |

AUC = area under the curve.

conditions associated with tissue necrosis such as colitis [37] or pleuronpneumonia [38] may have bolstered our hypothesis regarding the influence of age on the lack of utility of cfDNA in the critically-ill equine neonate.

The accuracy of fluorometry for the measurement of even small amounts of cfDNA has been previously demonstrated and validated [39]. While DNA extraction may improve DNA purity from samples, extraction processes have been shown to decrease total DNA concentration by removing small fragments [40–42] as reported in canine studies [43, 44]. Typical DNA extraction protocols are not sensitive when conserving small fragments <100 base pairs in size), yielding up to eight-fold variations in cfDNA concentrations [42]. Based on these data, cfDNA quantification of unaltered plasma appears acceptable [44] as we have done in this study. However, further investigation on the impact of DNA extraction on fluorometric cfDNA concentrations in equine plasma are needed.

The relatively small number of foals that died or were euthanized in this study compared to previous reports may also have affected our ability demonstrate the utility of cfDNA to predict outcome [3, 45]. In one published retrospective study analyzing equine neonatal admissions at a university teaching hospital between 1982 and 2008, 27.2% of foals did not survive to discharge [3]. A more recent retrospective of neonates admitted to a university or private referral hospital between 2008–2009 had a mortality rate of 21%. In this study, only 11/80 (14%) did not survive to discharge, limiting our ability to extrapolate the relationship between cell-free DNA levels and survival. Excluding healthy foals, 76% of sick non-septic and septic foals survived while 24% were euthanized or died. It is possible that the difference between studies may lie in differences in patient populations and diseases, and the large number of healthy foals enrolled in the study may have contributed to these data. Post-hoc analysis was performed to determine a sample size number where a clinically relevant difference (20%) in cfDNA between healthy and septic foals which found that 31 healthy and 31 septic foals would be needed. Further study with a larger population of critically ill foals may further define if cfDNA is associated with neonatal survival.

The definition of sepsis is a significant factor to consider in this study as well as past and future studies. This is particularly important in how foals are stratified and by what discriminating factors. Using the sepsis score and blood culture status to distinguish foals into categories (healthy, sick non-septic and sepsis) has inherent limitations and low reported sensitivities in neonatal foals [3, 7, 8, 20, 46]. False positive blood cultures have been documented in human literature and have been associated with increased hospital stays, inappropriate administration of antimicrobials and unnecessary hospital expense [47–49]. Blood cultures yielding non-pathogenic isolates can influence sepsis scoring systems and categorization schemes [48, 50, 51]. False negative blood cultures can also occur in septic foals for reasons such as the fastidious nature of some organisms that do not grow well in culture media or low circulating numbers of bacteria in the bloodstream at the time of sampling [3]. Prior administration of antimicrobials at the time of sample collection could also result in a negative culture result [36, 52], however guidelines in human medicine report that a delay in treatment is associated with increased morbidity and mortality [1, 53, 54]. The classification scheme used in this study has a sensitivity of 60% and specificity of 61% with an area under the curve of 0.71 [19]. The additional classification criteria of a positive blood culture possesses a documented sensitivity of only 61.9% in septic human neonates [6]. Given these findings, further study is needed to more accurately classify foals, particularly septic foals versus bacteremic foals.

In conclusion, plasma cell-free DNA concentration levels appear to be inadequate as a diagnostic biomarker for foals with sepsis or to predict survival in a neonatal population. Similar to studies in adult humans and dogs, the utility of cfDNA warrants further study in adult horses, notably horses with critical illness where cfDNA may have prognostic value in an

immunologically mature subject but is yet to be determined. Finally, the pursuit of a robust and accurate method to diagnose sepsis is still needed as illness severity scoring systems (e.g. sepsis score) and one-time blood culture methods appear woefully insensitive in this population.

## Supporting information

**S1 Checklist.**
(DOCX)

**S1 Data.**
(XLSX)

## Acknowledgments

The authors wish to thank the interns, residents and nurses at New Bolton Center for sample collection and their dedication to the management of the foals in this study.

## Author Contributions

**Conceptualization:** Samuel David Hurcombe.

**Data curation:** Sarah Florence Colmer, Samuel David Hurcombe.

**Formal analysis:** Sarah Florence Colmer, Darko Stefanovski, Samuel David Hurcombe.

**Funding acquisition:** Sarah Florence Colmer, Samuel David Hurcombe.

**Investigation:** Sarah Florence Colmer, Daniela Luethy, Michelle Abraham, Samuel David Hurcombe.

**Methodology:** Samuel David Hurcombe.

**Project administration:** Samuel David Hurcombe.

**Resources:** Samuel David Hurcombe.

**Supervision:** Samuel David Hurcombe.

**Writing – original draft:** Sarah Florence Colmer.

**Writing – review & editing:** Sarah Florence Colmer, Daniela Luethy, Michelle Abraham, Darko Stefanovski, Samuel David Hurcombe.

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
