## [Decision Letter · Decision Letter 0]

19 Jan 2021

PONE-D-20-34767

Utility of cell-free DNA concentrations and illness severity scores to predict survival in critically ill neonatal foals

PLOS ONE

Dear Dr. Hurcombe,

Thank you for submitting your manuscript to PLOS ONE. After careful consideration, we feel that it has merit but does not fully meet PLOS ONE’s publication criteria as it currently stands. Therefore, we invite you to submit a revised version of the manuscript that addresses the points raised during the review process.

We look forward to receiving your revised manuscript.

Kind regards,

Simon Clegg, PhD

Academic Editor

PLOS ONE

Additional Editor Comments:

Many thanks for submitting your manuscript to PLOS One

It was reviewed by two experts in the field, and they have recommended some modifications be made prior to acceptance.

I therefore invite you to make these changes and write a response to reviewers- this will greatly expedite review upon re-submission

I wish you the best of luck with your revisions

Hope you are keeping safe and well in these difficult times

Thanks

Simon

Reviewers' comments:

Reviewer's Responses to Questions

**Comments to the Author**

1. Is the manuscript technically sound, and do the data support the conclusions?

Reviewer #1: Yes

Reviewer #2: Partly

2. Has the statistical analysis been performed appropriately and rigorously? 

Reviewer #1: Yes

Reviewer #2: Yes

3. Have the authors made all data underlying the findings in their manuscript fully available?

Reviewer #1: Yes

Reviewer #2: Yes

4. Is the manuscript presented in an intelligible fashion and written in standard English?

Reviewer #1: Yes

Reviewer #2: Yes

5. Review Comments to the Author

Reviewer #1: Title: Utility of cell-free DNA concentrations and illness severity scores to predict survival in critically ill neonatal foals

#: PONE-D-20-34767

By: Colmer et al

Plasma cell-free DNA (cfDNA) levels have been associated with disease and survival in septic humans and dogs, but cfDNA studies in sick foals are lacking. The authors hypothesized that cfDNA would be detectable in plasma of sick foals, levels will be higher septic and sick-nonseptic compared to healthy foals, and increased cfDNA levels would be associated with non-survival. The team used healthy and sick foals to address the questions. Plasma cfDNA was detected in all foals, but no differences were found between healthy and sick foals. From the results it seems that cfDNA offers no clinical value to assess disease severity or likelihood of mortality in hospitalized foals. The proposal was well written, clear, with a clinical perspective. Concerns on this study relate group stratification/experimental design

Title: Minor comment: part of the title reads as if a goal was to use severity score to predict survival

Abstract: Clear and reflecting content of the manuscript. Issues to address relate to the methods/results

Introduction: OK – good justification is given

Methods: A concern in the methods relates to group stratification – age. For example, could the lack of difference between groups reflect the population age range? This could also relate to disease severity since older foals are usually not as sick as newborns. Therefore, youngest ones will have higher sepsis scores. Thus, the age factor could be confounding differences that can be addressed by restricting age to < 72 h. In addition, most hospitalized foals that are euthanized are < 72 h old.

Statistical analysis performed seems valid. Concerns relate to age and group stratification, which could influence interpretation

Results: As from previous comments, what was the median age/range at sampling? Was age statistically different between groups? This is not included in the study population (lines 188…) . If so, it could be a confounding factor on the statistics

The cluster of figure 2A indicates that there are no differences based on sepsis score, but how would this figure look if only septic (SS >12) are graphed? What is the correlation between cfDNA and SS in septic foals only? If most non-survivors were < 48 h (don’t know) and survivors >72 h, how would the analysis look in those of similar age (e.g. <96 h, and excluding healthy foals)? These questions will provide additional clarification on cfDNA and illness.

Did the team do some type of linearity /repeatability of this assay? Seems most samples were between 200-400 ng/ml regardless of disease severity. There are few studies evaluating this DNA measurement method in veterinary medicine, but values in septic dogs were much higher using 2 different methods (Letendre et al. 2017). Higher values and variability have been reported in septic people.

Line 193: In addition to total survival rate, list the survival rate for hospitalized/sick foals only. As presented is overestimating survival since all healthy survived – 69/80 vs 11/46

Were DNA concentrations higher in those with hypoperfusion/high lactate – leukopenia, independent of sepsis score?

Lines 198-201: Interesting that bacteria commonly isolated at referral centers (E. coli, Actinobacillus spp. were not isolated)

Discussion: Clear, addressing findings. However, I suggest to be cautious on the statement about lack of association between DNA and disease severity until further data analysis is carried out as suggested (age). Most studies in small animals and humans have shown a distinct association between SIRS/sepsis and cfDNA, with some even suggesting its clinical use. True that lower values have been found in neonates, which also appears to be the case for foals. Along the same line, perhaps the team should have included samples from healthy and sick horses with known cell injury (colitis). That would have supported the statement about cfDNA – neonate vs adult.

Good discussion about extracellular traps/NETs is provided and a concept rarely discussed in veterinary medicine.

Another confounding factor (limitation) is the number of non-surviving foals – bias from healthy foals and sample size. Line 321 doesn’t reflect a sick population but total population. Estimates about survival in hospitalized foals are given based on admission of sick ones (~50-60% survival rate).

Limitations of the sepsis score and blood culture are listed and valid.

The conclusion may need revision depending on further analysis as suggested.

References: OK

Figures: OK quality, however, as mentioned above, how would figure 2A look if only septic foals are plotted. Just an exercise.

Tables: OK

Reviewer #2: This manuscript addresses an interesting and relevant question -- if cell free DNA concentrations are impacted by sepsis and illness in neonatal foals, and if they are predictive of survival. The manuscript is well-written and the rationale for the study, the methods, and the data provided are generally clear. There are some important concerns with methodology and the presentation of the results, which are addressed in detail with regards to each section of the manuscript below.

ABSTRACT AND INTRODUCTION -- clear and well written, hypotheses and objectives clearly states

MATERIALS AND METHODS - generally clear and well organized. However, one major concern is the lack of validation of the cfDNA assay utilized in this study. No previous use of or validation of this assay on equine samples is cited, and no internal validation (positive and negative controls, accuracy, precision, intra-and inter-assay coefficients of variation) is provided. If this information can not be provided, it is impossible to know if the data and interpretations presented are valid. Also, the sample size calculations performed based on the prevalence of bacteremia in foals seems inadequate for the study's objectives. cfDNA concentrations from preliminary data in foals or studies in other species comparing septic and healthy individuals would have been more appropriate to base sample size calculations on, and the determine what % difference in cfDNA between septic and healthy foals might be clinically relevant. This study may be very underpowered -- the rationale for these sample size calculations should be better explained and justified.

RESULTS - This section could be better organized by division into sections with subheadings that discuss animal demographics, blood culture results, and outcome data separately to make it easier for the reader to follow. Additionally, much of the data provided in the text in the study population may be more efficiently provided in tabular form.

In table 1, the n provided for each group differ substantially from the group n provided in the study population text. This is concerning.

In general, the cfDNA data is provided in too many tables - as none of the differences between groups were significantly different, this may be better provided in a table or simply in the text with a smaller number of figures.

Failure to age-match foals between groups is substantial concern, as cfDNA concentrations in circulation in neonates may differ substantially with age. This should be addressed as a limitation and age distribution of the foals in each group should be provided. In lines 261-263, the authors mention that they further stratified the data by age, but it isn't clear if this is within each group of foals or in the total group, which could impact results if foals in different groups were different ages.

If Figure 4 is retained, individual animal data points should be shown as well, as in Figures 1-3.

Figures 5B and 5C are not necessary as the objective of this paper was not to assess the predictive ability of SS and NSIRS to predict survival. If those are removed, table 2 can and should also be removed and the cutoff value for cfDNA noted with Figure 5A.

DISCUSSION: Generally well written and clear, but study limitations, especially the lack of age-matching between groups, should be clearly addressed. The discussion of the limitations of the sepsis definition utilized is important but too lengthy in its current form (lines 327-353) -- this section should be edited to be more specific and concise.

The discussion of blood culture results in lines 353 - 363 is interesting but beyond the scope of this study. If retained, it should be shortened and refocused.

6. PLOS authors have the option to publish the peer review history of their article (what does this mean?). If published, this will include your full peer review and any attached files.

Reviewer #1: No

Reviewer #2: No

---

## [Author Response · Author response to Decision Letter 0]

9 Feb 2021

Response to Reviewers: PONE-D-20-34767

The authors wish to thank the reviewers for their thoughtful and thorough critique. We believe the edits to the manuscript significantly improve the quality of the study findings and we hope to have sufficiently address your concerns and questions. 

Sincerely,

The authors

Reviewer #1: Title: Utility of cell-free DNA concentrations and illness severity scores to predict survival in critically ill neonatal foals

#: PONE-D-20-34767

By: Colmer et al

Plasma cell-free DNA (cfDNA) levels have been associated with disease and survival in septic humans and dogs, but cfDNA studies in sick foals are lacking. The authors hypothesized that cfDNA would be detectable in plasma of sick foals, levels will be higher septic and sick-nonseptic compared to healthy foals, and increased cfDNA levels would be associated with non-survival. The team used healthy and sick foals to address the questions. Plasma cfDNA was detected in all foals, but no differences were found between healthy and sick foals. From the results it seems that cfDNA offers no clinical value to assess disease severity or likelihood of mortality in hospitalized foals. The proposal was well written, clear, with a clinical perspective. Concerns on this study relate group stratification/experimental design

Title: Minor comment: part of the title reads as if a goal was to use severity score to predict survival 

We did want to look at the ability of previously published illness scoring systems on their ability to predict survival, not just a diagnosis of sepsis. As such we would prefer to leave the title as is however defer to the editor to comment if this is unsatisfactory to the reviewer. 

Abstract: Clear and reflecting content of the manuscript. Issues to address relate to the methods/results

Introduction: OK – good justification is given

Methods: A concern in the methods relates to group stratification – age. For example, could the lack of difference between groups reflect the population age range? This could also relate to disease severity since older foals are usually not as sick as newborns. Therefore, youngest ones will have higher sepsis scores. Thus, the age factor could be confounding differences that can be addressed by restricting age to < 72 h. In addition, most hospitalized foals that are euthanized are < 72 h old.

We have further clarified this question in the manuscript (lines 198-202; line 208-209; lines 282-286, 325-329) and in more detail here. 

We found no correlation between age severity of disease as indicated by sepsis score (r=0.03, P=0.73)

spearman sepsisscore ageonintakedays

 Number of obs = 80

Spearman's rho = 0.0390

Test of Ho: sepsisscore and ageonintakedays are independent

 Prob > |t| = 0.7310

Further analysis adjusting for age, we found no association between sepsis score and cfDNA concentration. 

. reg sepsisscore ageonintakedays cfdnaconcmeanall3trials, robust

Linear regression Number of obs = 80

 F(2, 77) = 0.13

 Prob > F = 0.8752

 R-squared = 0.0017

 Root MSE = 5.8371

 | Robust

 sepsisscore | Coef. Std. Err. t P>|t| [95% Conf. Interval]

+----------------------------------------------------------------

 ageonintakedays | .088015 .2290484 0.38 0.702 -.3680785 .5441086

cfdnaconcmeanall3trials | .0017699 .0059873 0.30 0.768 -.0101523 .013692

 _cons | 6.291124 1.777412 3.54 0.001 2.751845 9.830404

Simple Spearman rank correlation and linear regression analysis: cfDNA and age

cfDNA and Age All Foals

Overall correlation r=0.1; P=0.26 (Spearman rank)

Linear regression P=0.366, goodness of fit R2 0.01, 95% CI of slope -6.1 to 16

Equation: cfDNA = 5.1 x age + 286 

Healthy Foals Only 

Overall correlation r=0.16, P=0.36 (Spearman rank)

Linear regression P=0.389, goodness of fit R2 = 0.022, 95%CI of slope -14 to 36

Equation: cfDNA = 11 x age + 272

Statistical analysis performed seems valid. Concerns relate to age and group stratification, which could influence interpretation

Results: As from previous comments, what was the median age/range at sampling? Was age statistically different between groups? This is not included in the study population (lines 188…) . If so, it could be a confounding factor on the statistics.

Please see above line references for details on age analysis. 

The cluster of figure 2A indicates that there are no differences based on sepsis score, but how would this figure look if only septic (SS >12) are graphed? What is the correlation between cfDNA and SS in septic foals only? If most non-survivors were < 48 h (don’t know) and survivors >72 h, how would the analysis look in those of similar age (e.g. <96 h, and excluding healthy foals)? These questions will provide additional clarification on cfDNA and illness.

There was no difference in age between survivors and non-survivors in this population (P=0.47; line 208).

Here are the median and mean ages for each of the classification groups and also the P values. 

. table caseoutcomedischargevsdeath, c(median ageonintakedays iqr ageonint

> akedays count ageonintakedays)

Case |

outcome |

(discharg |

e vs |

death) | med(ageoni~s) iqr(ageoni~s) N(ageoni~s)

----------+--------------------------------------------

 0 | 0 1 69

 1 | 0 3 11

. table caseoutcomedischargevsdeath, c(mean ageonintakedays sd ageonintake

> days count ageonintakedays)

Case |

outcome |

(discharg |

e vs |

death) | mean(ageoni~s) sd(ageoni~s) N(ageoni~s)

----------+-----------------------------------------------

 0 | .81884058 1.557645 69

 1 | 1.6818182 3.002272 11

. kwallis ageonintakedays, by(caseoutcome)

Kruskal-Wallis equality-of-populations rank test

 +---------------------------+

 | caseou~h | Obs | Rank Sum |

 |----------+-----+----------|

 | 0 | 69 | 2743.00 |

 | 1 | 11 | 497.00 |

 +---------------------------+

chi-squared = 0.518 with 1 d.f.

probability = 0.4718

chi-squared with ties = 0.677 with 1 d.f.

probability = 0.4106

Here is the cfDNA concentrations for foals with SS >12 only. There was no correlation between SS and cfDNA for foals when assessing foals with SS >/= 12 (Line 255-260; Table 1).

Did the team do some type of linearity /repeatability of this assay? Seems most samples were between 200-400 ng/ml regardless of disease severity. There are few studies evaluating this DNA measurement method in veterinary medicine, but values in septic dogs were much higher using 2 different methods (Letendre et al. 2017). Higher values and variability have been reported in septic people.

The authors recreated figure 2A above, such that septic foals >= 12 are graphed (not largely different from the original figure 2A in the manuscript). The correlation between cfDNA and SS in septic foals only is not significant with a P value of 0.3593 

The authors recognize the decreased overall numbers in this study (generally <400 ng/mL) and discuss some plausible reasons for the general decrease compared to other studies in veterinary literature such as that described by Letendre et al. 2017 including age and immunological naivete. Regarding repeatability, the assays were run in triplicate, using the average value of the three trials for each sample. Further assay specific details have been added (Lines 145-155). 

Line 193: In addition to total survival rate, list the survival rate for hospitalized/sick foals only. As presented is overestimating survival since all healthy survived – 69/80 vs 11/46

Were DNA concentrations higher in those with hypoperfusion/high lactate – leukopenia, independent of sepsis score?

The authors have now included survival rate for sick non-septic and septic foals specifically: a total of 35/46 (76%) of sick non-septic or septic foals survived to discharge, while they exhibited a mortality rate of 24% (11/46 euthanized or died). 

There was no correlation with cfDNA and neutrophil count (r=0.2, P=0.08), total leukocyte count (r=0.18, P=0.12) or L-lactate concentration (r=-0.12, P=0.29). Further linear regression analysis also did not show any strong relationship between cfDNA and these variables. We have elected not to include these data in the manuscript however if the reviewer and/or editor feels it is additive/useful to the manuscript, we can. 

Lines 198-201: Interesting that bacteria commonly isolated at referral centers (E. coli, Actinobacillus spp. were not isolated)

Authors agree that this was an interesting finding, although E. coli was one of the most common isolates obtained overall as opposed to strictly the population that died.

Discussion: Clear, addressing findings. However, I suggest to be cautious on the statement about lack of association between DNA and disease severity until further data analysis is carried out as suggested (age). Most studies in small animals and humans have shown a distinct association between SIRS/sepsis and cfDNA, with some even suggesting its clinical use. True that lower values have been found in neonates, which also appears to be the case for foals. Along the same line, perhaps the team should have included samples from healthy and sick horses with known cell injury (colitis). That would have supported the statement about cfDNA – neonate vs adult.

We agree. The apparent effect of age was not something expected by the team, especially now after further age specific analysis. Sampling horses with and without critical illness as well as following cfDNA concentrations from birth to adulthood would be very interesting to study in a prospective manner in the future. Our speculations about neonate vs adult are, as discussed, based in literature in other species, but remain to be elucidated. 

Good discussion about extracellular traps/NETs is provided and a concept rarely discussed in veterinary medicine.

Another confounding factor (limitation) is the number of non-surviving foals – bias from healthy foals and sample size. Line 321 doesn’t reflect a sick population but total population. Estimates about survival in hospitalized foals are given based on admission of sick ones (~50-60% survival rate).

We agree. The lower mortality rate in this population compared to previous reports is an important factor to bear in mind when making conclusions about the data. We have been deliberate to address this in the discussion (Lines 354-362). 

Limitations of the sepsis score and blood culture are listed and valid.

The conclusion may need revision depending on further analysis as suggested.

References: OK

Figures: OK quality, however, as mentioned above, how would figure 2A look if only septic foals are plotted. Just an exercise.

Tables: OK

Reviewer #2: This manuscript addresses an interesting and relevant question -- if cell free DNA concentrations are impacted by sepsis and illness in neonatal foals, and if they are predictive of survival. The manuscript is well-written and the rationale for the study, the methods, and the data provided are generally clear. There are some important concerns with methodology and the presentation of the results, which are addressed in detail with regards to each section of the manuscript below.

ABSTRACT AND INTRODUCTION -- clear and well written, hypotheses and objectives clearly states

MATERIALS AND METHODS - generally clear and well organized. However, one major concern is the lack of validation of the cfDNA assay utilized in this study. No previous use of or validation of this assay on equine samples is cited, and no internal validation (positive and negative controls, accuracy, precision, intra-and inter-assay coefficients of variation) is provided. If this information can not be provided, it is impossible to know if the data and interpretations presented are valid. 

Fair point and we have added details on the assay used, pilot data and intra-/interassay CV values. Lines 145-155.

Moreover, we have added further discussion about the influence of DNA extraction vs no extraction as well as DNA quantification methodology (Lines 339-348)

Also, the sample size calculations performed based on the prevalence of bacteremia in foals seems inadequate for the study's objectives. cfDNA concentrations from preliminary data in foals or studies in other species comparing septic and healthy individuals would have been more appropriate to base sample size calculations on, and the determine what % difference in cfDNA between septic and healthy foals might be clinically relevant. This study may be very underpowered -- the rationale for these sample size calculations should be better explained and justified.

We have added further details to the methods and discussion regarding sample size calculation and also performed a post-hoc analysis (Lines 179-182; 359-361). 

Post-hoc power analysis to identify the number of animals required to identify 3% increase in cfDNA between healthy and septic animals. Assuming power=0.8 and alpha=0.05, the resulting samples size is N=2380 animals or n=1190. Unfortunately, this number is unattainable. Furthermore, we do not think that 3% change is clinically relevant change. However, with a more realistic 20% change in cfDNA concentration between septic and healthy animals, we find the total sample to be N=62, or per group that is n=31. Since we did have sufficient numbers in our cohort we conclude that this study was correctly powered. 

We used bacteremia prevalence rates to determine sample size as there has not been cfDNA has not been evaluated in horses, notably critically ill horses. Increased cfDNA concentrations have been observed in humans with bacteremia (Forsblom et al; Huttunen et al) and bacteremia is a commonly used inclusion criteria along with clinical findings and/or sepsis score for the diagnosis of sepsis in equine neonates. 

Forsblom E, Aittoniemi J, Ruostalainen E, et al. High cell-free DNA predicts fatal outcome among Staphylococcus aureus bacteraemia patients with intensive care unit treatment. PLoS One 2014;9(2):e87741

Huttunen R, Kuparinen T, Jylhava J, et al. Fatal outcome in bacteremia is characterized by high plasma cell free DNA concentration and apoptotic DNA fragmentation: a prospective cohort study. PLoS One 2011;6(7):e21700

RESULTS - This section could be better organized by division into sections with subheadings that discuss animal demographics, blood culture results, and outcome data separately to make it easier for the reader to follow. Additionally, much of the data provided in the text in the study population may be more efficiently provided in tabular form.

The authors have created “study population demographics, blood culture results, cfDNA concentrations and age” subheadings to aid in organization. 

In table 1, the n provided for each group differ substantially from the group n provided in the study population text. This is concerning.

Respectfully, we have specified that the n in table 1 are the numbers of foals classified by sepsis score alone (excluding blood culture status; Lines 255-260). We have further clarified this in the table 1 legend.

In general, the cfDNA data is provided in too many tables - as none of the differences between groups were significantly different, this may be better provided in a table or simply in the text with a smaller number of figures.

Failure to age-match foals between groups is substantial concern, as cfDNA concentrations in circulation in neonates may differ substantially with age. 

As discussed above with reviewer 1, we have added further analysis and details on age. 

lines 198-202; line 208-209; lines 282-286, 325-329

. spearman cfdnaconcmeanbothtrials age

 Number of obs = 69

Spearman's rho = -0.0272

Test of Ho: cfdnaconcmeanbotht~s and ageonintakedays are independent

 Prob > |t| = 0.8247

This should be addressed as a limitation and age distribution of the foals in each group should be provided. In lines 261-263, the authors mention that they further stratified the data by age, but it isn't clear if this is within each group of foals or in the total group, which could impact results if foals in different groups were different ages.

If Figure 4 is retained, individual animal data points should be shown as well, as in Figures 1-3.

Figure 4 has been reworked.

Figures 5B and 5C are not necessary as the objective of this paper was not to assess the predictive ability of SS and NSIRS to predict survival. If those are removed, table 2 can and should also be removed and the cutoff value for cfDNA noted with Figure 5A.

Respectfully, we have further clarified our objectives in the introduction and believe these data are an important part of the manuscript. 

DISCUSSION: Generally well written and clear, but study limitations, especially the lack of age-matching between groups, should be clearly addressed. The discussion of the limitations of the sepsis definition utilized is important but too lengthy in its current form (lines 327-353) -- this section should be edited to be more specific and concise.

The authors agree that the discussion of limitations of the sepsis definition was too lengthy in its previous form. It has been edited to be more specific, focused and concise (29 lines down to 19 lines).

The discussion of blood culture results in lines 353 - 363 is interesting but beyond the scope of this study. If retained, it should be shortened and refocused.

The authors agree that, while interesting, this is in fact beyond the scope of the research question and relevant results. This paragraph of the discussion has been removed.

---

## [Decision Letter · Decision Letter 1]

6 Apr 2021

PONE-D-20-34767R1

Utility of cell-free DNA concentrations and illness severity scores to predict survival in critically ill neonatal foals

PLOS ONE

Dear Dr. Hurcombe,

Thank you for submitting your manuscript to PLOS ONE. After careful consideration, we feel that it has merit but does not fully meet PLOS ONE’s publication criteria as it currently stands. Therefore, we invite you to submit a revised version of the manuscript that addresses the points raised during the review process.

Many thanks for submitting your manuscript to PLOS One

It was reviewed by two experts in the field, and they have recommended some minor modifications be made prior to acceptance

I therefore invite you to make these changes and to write a response to reviewers which will expedite revision upon resubmission

I wish you the best of luck with your modifications

Hope you are keeping safe and well in these difficult times

Thanks

Simon

We look forward to receiving your revised manuscript.

Kind regards,

Simon Clegg, PhD

Academic Editor

PLOS ONE

Journal Requirements:

Reviewers' comments:

Reviewer's Responses to Questions

**Comments to the Author**

1. If the authors have adequately addressed your comments raised in a previous round of review and you feel that this manuscript is now acceptable for publication, you may indicate that here to bypass the “Comments to the Author” section, enter your conflict of interest statement in the “Confidential to Editor” section, and submit your "Accept" recommendation.

Reviewer #1: All comments have been addressed

Reviewer #3: All comments have been addressed

2. Is the manuscript technically sound, and do the data support the conclusions?

Reviewer #1: Yes

Reviewer #3: Yes

3. Has the statistical analysis been performed appropriately and rigorously? 

Reviewer #1: Yes

Reviewer #3: Yes

4. Have the authors made all data underlying the findings in their manuscript fully available?

Reviewer #1: Yes

Reviewer #3: Yes

5. Is the manuscript presented in an intelligible fashion and written in standard English?

Reviewer #1: Yes

Reviewer #3: Yes

6. Review Comments to the Author

Reviewer #1: Title: Utility of cell-free DNA concentrations and illness severity scores to predict survival in critically ill neonatal foals

#: PONE-D-20-34767R1

By: Colmer et al

The authors addressed major concerns raised in the initial submission of this manuscript. Other than minor grammatical errors or typos this reviewer is pleased with the response.

Minor:

While some findings remain intriguing, in particular the lack of statistical differences among groups, assuming that cell lysis from SIRS occurred in these foals, results could be real, supporting that cfDNA has limited use as an indicator of disease severity in critically ill equine neonates. Unfortunately, and as discussed, including animals with well documented tissue necrosis (e.g. colitis, pleuropneumonia, etc) as potential positive controls would have strengthen this manuscript (limitations).

It is unclear in the tables whether the reported SE is of the mean or the median (not the same). Clarification is important as SE of the median is often invalid because it makes assumption of normality (unless that information was from a normally distributed data set).

Reviewer #3: All the comments have been addressed and the manuscript is a welcome addition to the literature in this area. I have five very minor grammatical and typo comments below but these can almost be discarded. I congratulate the authors on a very nice manuscript.

Line 45- Slightly difficult to read. Maybe a comma around however may help?

Line 94- maybe saying 10 days old may make this clearer?

Line 171- Maybe reword to ‘by allowing them to come to…’

Line 181- change ares to are

Line 241- bacteremic is spelt incorrectly

7. PLOS authors have the option to publish the peer review history of their article (what does this mean?). If published, this will include your full peer review and any attached files.

Reviewer #1: No

Reviewer #3: No

---

## [Author Response · Author response to Decision Letter 1]

9 Apr 2021

Response to Reviewers: PONE-D-20-34767R1

The authors wish to once again thank the reviewers for the time and expertise that went into their critique. We believe the most recent edits to the manuscript have further improved the quality of the report and addressed your concerns. 

Sincerely,

The authors

Reviewer #1: Title: Utility of cell-free DNA concentrations and illness severity scores to predict survival in critically ill neonatal foals

#: PONE-D-20-34767R1

By: Colmer et al

The authors addressed major concerns raised in the initial submission of this manuscript. Other than minor grammatical errors or typos this reviewer is pleased with the response.

Minor:

While some findings remain intriguing, in particular the lack of statistical differences among groups, assuming that cell lysis from SIRS occurred in these foals, results could be real, supporting that cfDNA has limited use as an indicator of disease severity in critically ill equine neonates. Unfortunately, and as discussed, including animals with well documented tissue necrosis (e.g. colitis, pleuropneumonia, etc) as potential positive controls would have strengthen this manuscript (limitations).

The authors agree that including animals with well-documented tissue necrosis (colitis or pleuropneumonia, etc) as positive controls could have strengthened the theory behind age influencing the lack of utility of cfDNA in the equine neonate discussed in this manuscript (line 341). 

It is unclear in the tables whether the reported SE is of the mean or the median (not the same). Clarification is important as SE of the median is often invalid because it makes assumption of normality (unless that information was from a normally distributed data set).

We have clarified the data distribution and edited the table to show median and range. 

Reviewer #3: All the comments have been addressed and the manuscript is a welcome addition to the literature in this area. I have five very minor grammatical and typo comments below but these can almost be discarded. I congratulate the authors on a very nice manuscript.

Line 45- Slightly difficult to read. Maybe a comma around however may help?

Line 94- maybe saying 10 days old may make this clearer?

Line 171- Maybe reword to ‘by allowing them to come to…’

Line 181- change ares to are

Line 241- bacteremic is spelt incorrectly

The authors appreciate and acknowledge the aforementioned suggestions in grammar and spelling, and the suggested alterations have been made.

---

## [Editor Report · Decision Letter 2]

13 Apr 2021

Utility of cell-free DNA concentrations and illness severity scores to predict survival in critically ill neonatal foals

PONE-D-20-34767R2

Dear Dr. Hurcombe,

We’re pleased to inform you that your manuscript has been judged scientifically suitable for publication and will be formally accepted for publication once it meets all outstanding technical requirements.

Kind regards,

Simon Clegg, PhD

Academic Editor

PLOS ONE

Additional Editor Comments:

Many thanks for resubmitting your manuscript to PLOS One

As you have addressed all the comments and the manuscript reads well, I have recommended it for publication

You should hear from the Editorial Office shortly.

It was a pleasure working with you and I wish you the best of luck for your future research

Hope you are keeping safe and well in these difficult times

Thanks

Simon

---

## [Editor Report · Acceptance letter]

16 Apr 2021

PONE-D-20-34767R2 

Utility of cell-free DNA concentrations and illness severity scores to predict survival in critically ill neonatal foals 

Dear Dr. Hurcombe:

I'm pleased to inform you that your manuscript has been deemed suitable for publication in PLOS ONE. Congratulations! Your manuscript is now with our production department. 

Kind regards, 

on behalf of

Dr. Simon Clegg 

Academic Editor

PLOS ONE